# Copper catalyzed late-stage C(sp$^3$)-H functionalization of nitrogen heterocycles

Zhe Chang[1], Jialin Huang[1], Si Wang[1], Geshuyi Chen[2], Heng Zhao[1], Rui Wang [2✉] & Depeng Zhao [1✉]

Nitrogen heterocycle represents a ubiquitous skeleton in natural products and drugs. Late-stage C(sp$^3$)-H bond functionalization of N-heterocycles with broad substrate scope remains a challenge and of particular significance to modern chemical synthesis and pharmaceutical chemistry. Here, we demonstrate copper-catalysed late-stage C(sp$^3$)-H functionalizaion of N-heterocycles using commercially available catalysts under mild reaction conditions. We have investigated 8 types of N-heterocycles which are usually found as medicinally important skeletons. The scope and utility of this approach are demonstrated by late-stage C(sp$^3$)-H modification of these heterocycles including a number of pharmaceuticals with a broad range of nucleophiles, e.g. methylation, arylation, azidination, mono-deuteration and glycoconjugation etc. Preliminary mechanistic studies reveal that the reaction undergoes a C-H fluorination process which is followed by a nucleophilic substitution.

[1] Guangdong Provincial Key Laboratory of Chiral Molecule and Drug Discovery, School of Pharmaceutical Sciences, Sun Yat-Sen University, Guangzhou, China. [2] Key Laboratory of Preclinical Study for New Drugs of Gansu Province, School of Basic Medical Sciences, Lanzhou University, Lanzhou, China. ✉email: wangrui@lzu.edu.cn; zhaodp5@mail.sysu.edu.cn

Nitrogen heterocycles (N-heterocycles) hold a special position in natural products chemistry and medicinal chemistry as evidenced by that ~60% of US Food and Drug Administration-approved drugs contain a N-heterocycle[1, 2]. Direct C(sp³)–H bond functionalization of N-heterocycles represents an attractive topic and is of paramount importance to drug development[3–9]. A number of approaches on C–H bond alkylation of N-heterocycle have been developed so far, e.g., C–H oxidation to iminium intermediate[6–8, 10–13], lithiation of α-position, C(sp³)–H bond activation[14–17], photoredox-based approaches, etc[18–24]. In particular, cross-dehydrogenative coupling (CDC), which proceeds via C–H oxidation to iminium intermediate, developed by Li[25, 26] based on the oxidative activation of C(sp³)–H bonds of amines and ethers has been intensively studied (Fig. 1b)[25, 26]. However, the scope of the amine coupling partner is limited to a few electron-rich aromatic amines. An elegant C–H alkylation of cyclic amines was developed recently

by Seidel and colleagues[14, 15, 27] using organolithium as a strong base and ketones as a hydride acceptor to generate cyclic imine, which was subsequently captured by organolithium and Grignard reagent. However, the use of organolithium restricts its application in late-stage functionalization. Despite all these advances, most of these methods developed so far have focused on simple azacycles, and late-stage C(sp³)–H bond functionalization of N-heterocycles with broad substrate scope remains a challenge. Late-stage C(sp³)–H hydroxylation of N-heterocycles furnishing hemiaminal intermediate, which undergoes further iminium formation and alkylation, offers a general approach to access various functionalization. However, no general approach has been developed until recently. Traditionally, hemiaminal were prepared either from electrochemical oxidation of amine or from reduction of lactams[13, 28–32]. The challenge in direct C(sp³)–H hydroxylation probably lies in over-oxidation to carbonyl or elimination to olefin at high temperature or under forcing

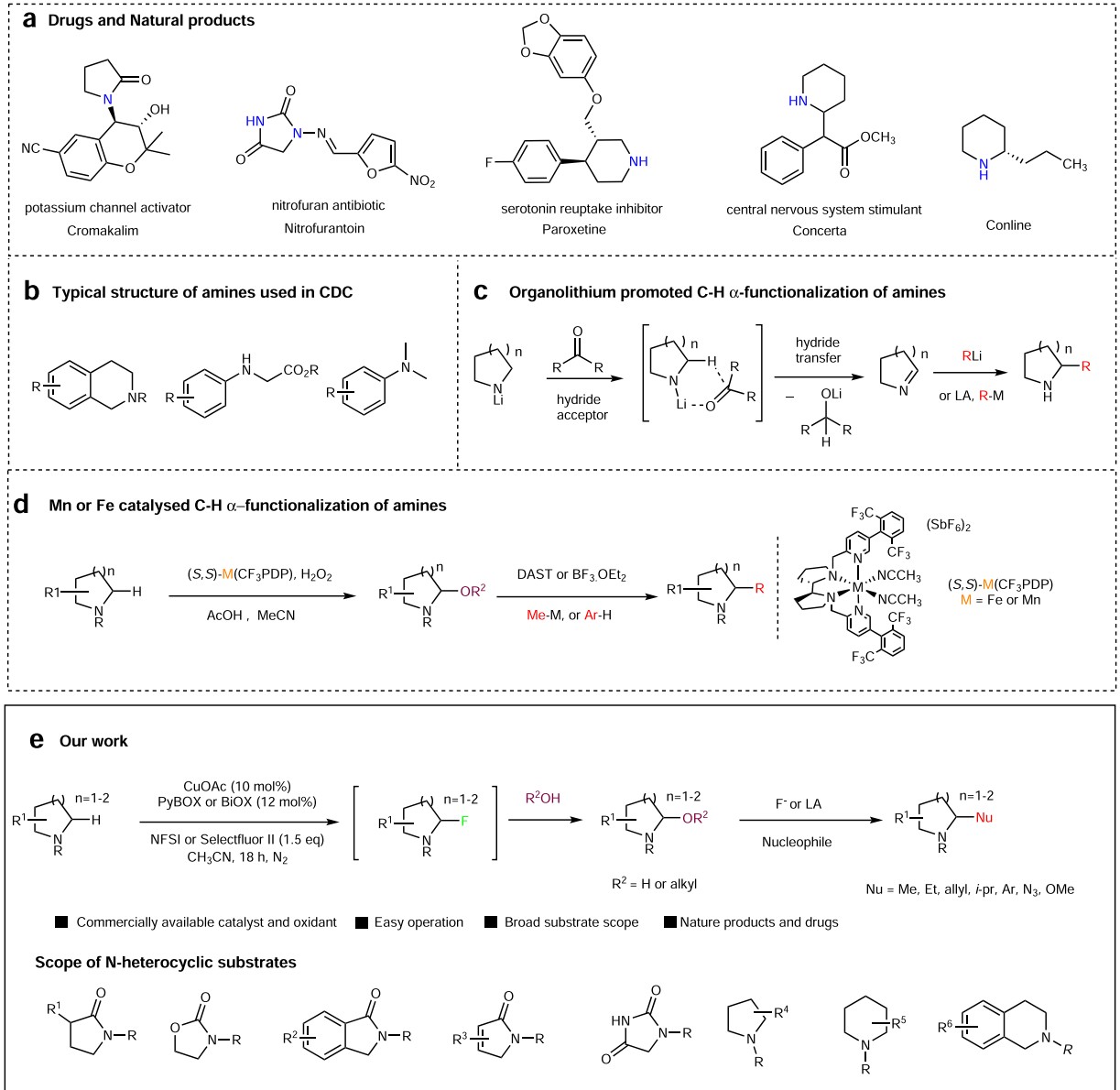

**Fig. 1 C(sp³)–H functionalization of N-heterocycles. a** Representative examples of N-heterocycles in pharmaceuticals and natural products. **b** Previously reported nitrogenous substrates of CDC reactions. **c** Previous work on organolithium promoted C–H α-functionalization of amines. **d** Previous work on Mn- or Fe-catalyzed C–H α-functionalization of amines. **e** This work: Cu-catalyzed C–H functionalization of N-heterocycles with NFSI or Selectflour II as oxidants and scope of N-heterocyclic substrates.

conditions[33, 34]. During the course of our research, White and colleagues[8] reported a Mn-catalyzed C($sp^3$)–H hydroxylation of N-heterocycles, including many drugs and the hemiaminal intermediates, was transformed further to methylated products. The same group earlier reported an Fe-catalyzed C($sp^3$)–H hydroxylation of N-heterocycles[35]. However, in these two cases, the catalyst employed is not commercially available, which requires ten steps to complete the synthesis and great care should be taken to avoid over-oxidation, e.g., dropwise addition of $H_2O_2$ using a syringe pump and low reaction temperature.

Herein, we show a Cu(I)-catalyzed late-stage C($sp^3$)–H functionalization of N-heterocycles via CDC approach using $H_2O$ or alcohol as coupling partners. The hydroxylation/alkoxylation of a variety of N-heterocycles proceeds in good yield under mild reaction conditions. The resulting hemiaminal or aminal intermediates were further functionalized with a broad range of nucleophiles promoted by Lewis acid or fluorination reagent. Compared with the previous work, our system has the following advantages: (1) the BOX type ligands employed in our approach are commercial; (2) common functional groups, such as C=C double bonds and secondary alcohols, which are not suitable for previous approach, are well compatible with our system; (3) a radical fluorination process is involved in this approach and the fluorinated intermediates have the potential to be transformed in situ to many other functional groups in a single step, whereas Mn/$H_2O_2$ chemistry is only a radical hydroxylation.

## Results

**Reaction optimization**. Inspired by recent advances in copper-catalyzed radical C–H activation using N–F reagents[36–39], in particular C–H fluorination[40], we reasoned if C–H functionalization of α-position of nitrogen could be achieved via copper-catalyzed C–H fluorination, due to the similar bond dissociation energy (BDE) of benzylic C–H bonds and α(C–H) bond of N-heterocycles[41, 42]. The resulting fluorinated intermediate is prone to hydrolysis under acidic conditions to give hemiaminal. This approach is advantageous over direct hydroxylation using OH radical, as hyperconjugative activation of hemiaminals often leads to over-oxidation to the carbonyl[43–45].

To test our hypothesis, we began our study by the reaction of oxazolidinones **1a** and $H_2O$ in the presence of 10 mol% of CuOTf/**L1** as the catalyst and N-fluorobenzenesulfonimide (NFSI) (1.5 eq.) as the oxidant in $CH_3CN$ at 25 °C (Table 1, entry 1). Gratifyingly, the desired CDC product **2a** was obtained in 26% yield albeit with low conversion. Encouraged by this result, we further systematically optimized the reaction conditions to improve the yield of the reaction. When the reaction was performed without a ligand, the reaction was sluggish and only trace amount of **2a** was produced (entry 2). A survey of other solvents was then carried out (entries 3–5), but no one led to a higher yield than in $CH_3CN$. When the reaction was performed at elevated temperature of 35 °C, a slightly higher yield of **2a** was achieved without byproduct formation from elimination or over-oxidation (40% yield, entry 6). Then we turned our attention to Cu salts. Cu(II) catalyst generated from $Cu(OTf)_2$ and **L1** was less efficient, and failed to give a higher yield (entry 7). The yield was further improved to 73% by using CuBr in place of CuOTf (entry 8). To our delight, with the catalyst formed from CuOAc and **L1**, the yield of **2a** was dramatically improved to 89% (entry 9). Screening of other PyBOX ligands showed that the substituent at the stereogenic center was crucial to the efficiency of the reaction (entries 9–12). Catalysts formed from **L1** or **L2** with aromatic rings gave significantly higher yields than **L3** or **L4** with aliphatic substituents. We also expanded the application of this C–H functionalization reaction toward the synthesis of aminals. When

MeOH was used as the coupling partner, the desired product **2aa** was successfully obtained in 95% yield (Table 1, entry 17). Controlled experiments were performed and indicated that the reaction did not proceed without Cu salts or NFSI (Supplementary Table 5).

**Substrate scope evaluation**. After optimizing the reaction conditions, we turned to explore the scope of N-heterocyclic substrates of this reaction (Fig. 2). Overall, despite the wide range of C–H BDEs of the N-heterocycles, we successfully applied this protocol to 34 compounds including 8 different types of N-heterocycles, which were usually found as important skeletons in pharmaceuticals and natural products. By changing different ligands and N–F reagents, all the N-heterocycles investigated were applicable to this strategy affording the desired products in moderate-to-good yields (41–97%). The reactions of oxazolidinones and isoindolinones derivatives proceeded smoothly under the standard reaction conditions with **L1**/CuOAc/NFSI. For the oxazolidinones, a higher yield was found when *para*-CN (**2a**, 89%) rather than *meta*-CN (**2b**, 77%) was used as a substituent group on the phenyl ring. The yield of *para*-Cl substituted **2c** is lower than that of *para*-CN substituted **2a** (56% vs. 89%). Due to the fact that hemiaminal products and the substrates with N-imine substituent were difficult to separate, the corresponding aminal products **2d**–**2h** were reported instead (85–97% yield). In the case of isoindolinone, aminal products were given instead as well, due to solubility issue of the corresponding hemiaminals. The CDC products **2i**–**2m** were all successfully formed in high yields (80–92%). Isindolinone **2l** with electron-rich substituent on the benzene ring gave a higher yield than other isoindolinone substrates (92% vs. 80–85%). It is worth mentioning that lactams gave low yields under the standard reaction conditions. Slightly modified conditions were applied to lactams, e.g., CuOAc/**L2**/20 °C, to avoid the formation of C–C double bond by elimination or the carbonyl by over-oxidation (**2n**–**2o**, 72–79% yield). Pyrrolones N-heterocycle, in which low reaction efficiency was observed using CuOAc/**L1**, also proved to be good substrates with CuOAc/**L2**, affording products **2p**–**2r** in good yields (66% and 59%, respectively).

For the common N-heterocycle pyrrolidines, initially, we studied the N-protecting groups (Supplementary Table 1) and nitrobenzenesulfonyl (Ns) was eventually identified as the optimal N-protecting group for this hydroxylation reaction, which was in accordance with previous reports[8, 35]. A survey of ligands (Supplementary Table 2), F-reagent (Supplementary Table 4), and temperature indicated that the optimal reaction conditions were CuOAc/**L6**/Selectfluor II/20 °C. Under these optimized conditions, Ns-pyrrolidine **1s** and Ns-pyrrolidines bearing more activated tertiary C–H bonds could be tolerated to afford the target products in good yields with high regioselectivities (**2s**–**2v**, 51–73% yield). In the case of Ns-protected piperidine, relatively harsh conditions were required (CuOAc (0.2 eq), ligand **L6** (0.22 eq), Selectflour II (2.0 eq)) to achieve reasonable yield (**2w**, 41%). Moreover, the scope of this protocol was successfully extended to Ns-tetrahydroisoquinolines delivering desired products **2x** and **2y** in moderate yields (52% and 65%, respectively).

Next, we evaluated the performance of this C–H functionalization reaction with various alcohols. Oxazolidinone **1d** was selected as a representative for these studies. Several alcohols, including ethanol, isopropanol, *n*-butyl alcohol, and cyclohexanol were applicable to this new protocol affording the CDC products in excellent yields (**2da**–**2dd**, 76–85%). It should be mentioned that benzyl alcohol with benzylic C–H bond and 5-hexen-1-ol with a terminal double bond were compatible to the reaction

**Table 1 Reaction optimization[a].**

| Entry | [Cu] | Solvent | T (°C) | Ligand | Product | Yield[b] |
|---|---|---|---|---|---|---|
| 1 | CuOTf | CH$_3$CN | 25 | L1 | 2a | 26% |
| 2 | CuOTf | CH$_3$CN | 25 | - | 2a | Trace |
| 3 | CuOTf | DCM | 25 | L1 | 2a | 5% |
| 4 | CuOTf | DCE | 25 | L1 | 2a | Trace |
| 5 | CuOTf | acetone | 25 | L1 | 2a | 11% |
| 6 | CuOTf | CH$_3$CN | 35 | L1 | 2a | 40% |
| 7 | Cu(OTf)$_2$ | CH$_3$CN | 35 | L1 | 2a | 29% |
| 8 | CuBr | CH$_3$CN | 35 | L1 | 2a | 73% |
| 9 | CuOAc | CH$_3$CN | 35 | L1 | 2a | 89% |
| 10 | CuOAc | CH$_3$CN | 35 | L2 | 2a | 78% |
| 11 | CuOAc | CH$_3$CN | 35 | L3 | 2a | Trace |
| 12 | CuOAc | CH$_3$CN | 35 | L4 | 2a | 31% |
| 14 | CuOAc | CH$_3$CN | 35 | L5 | 2a | 78% |
| 15 | CuOAc | CH$_3$CN | 35 | L6 | 2a | 48% |
| 16 | CuOAc | CH$_3$CN | 35 | L7 | 2a | 40% |
| 17[c] | CuOAc | CH$_3$CN | 35 | L1 | 2aa | 95% |

[a]Reaction conditions: 1a (0.2 mmol), H$_2$O (0.6 mmol), CuOAc (0.02 mmol), ligand (0.024 mmol), NFSI (0.3 mmol), and solvent (2.0 mL), 18 h.
[b]Isolated yield.
[c]Reaction conditions: 1a (0.2 mmol), H$_2$O (0.6 mmol), MeOH (0.6 mmol), CuOAc (0.02 mmol), NFSI (0.3 mmol), L1 (0.024 mmol), CH$_3$CN (2.0 mL), 18 h.

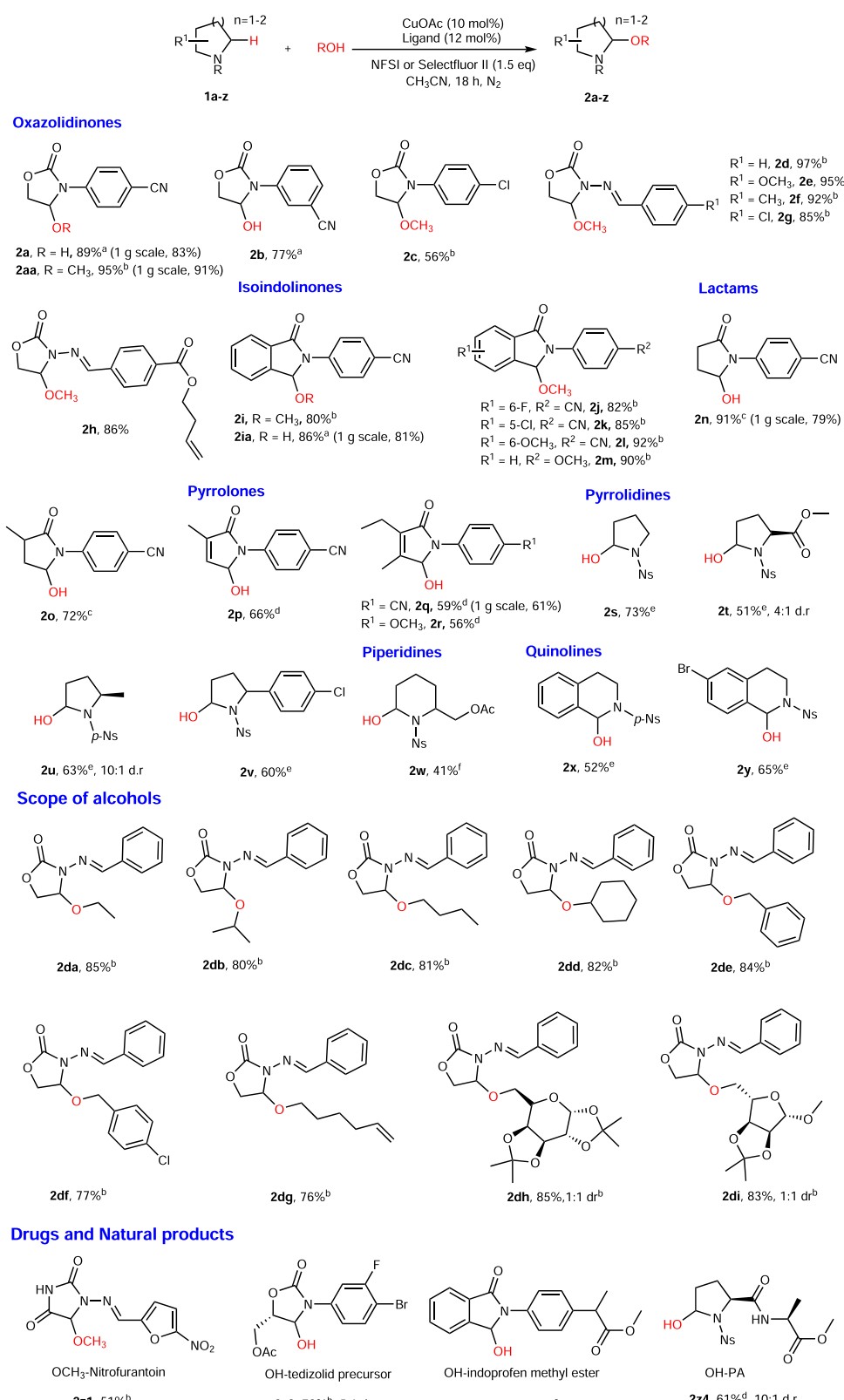

**Fig. 2 Scope of N-heterocyclic substrates.** Reaction conditions: N-heterocyclic substrates (0.2 mmol), $H_2O$ (0.6 mmol), CuOAc (0.02 mmol), ligand (0.024 mmol), NFSI (0.3 mmol), $CH_3CN$ (2.0 mL), under $N_2$; isolated yields reported. [a]**L1**, temperature: 35 °C. [b]**L1**, ROH (0.6 mmol). [c]**L2**, temperature: 20 °C. [d]Ligand **L2**, temperature: 30 °C. [e]**L6**, Selectflour II instead of NFSI, temperature: 20 °C. [f]CuOAc (0.2 eq.), **L6** (0.22 eq), Selectflour II (2.0 eq.) instead of NFSI, temperature: 20 °C.

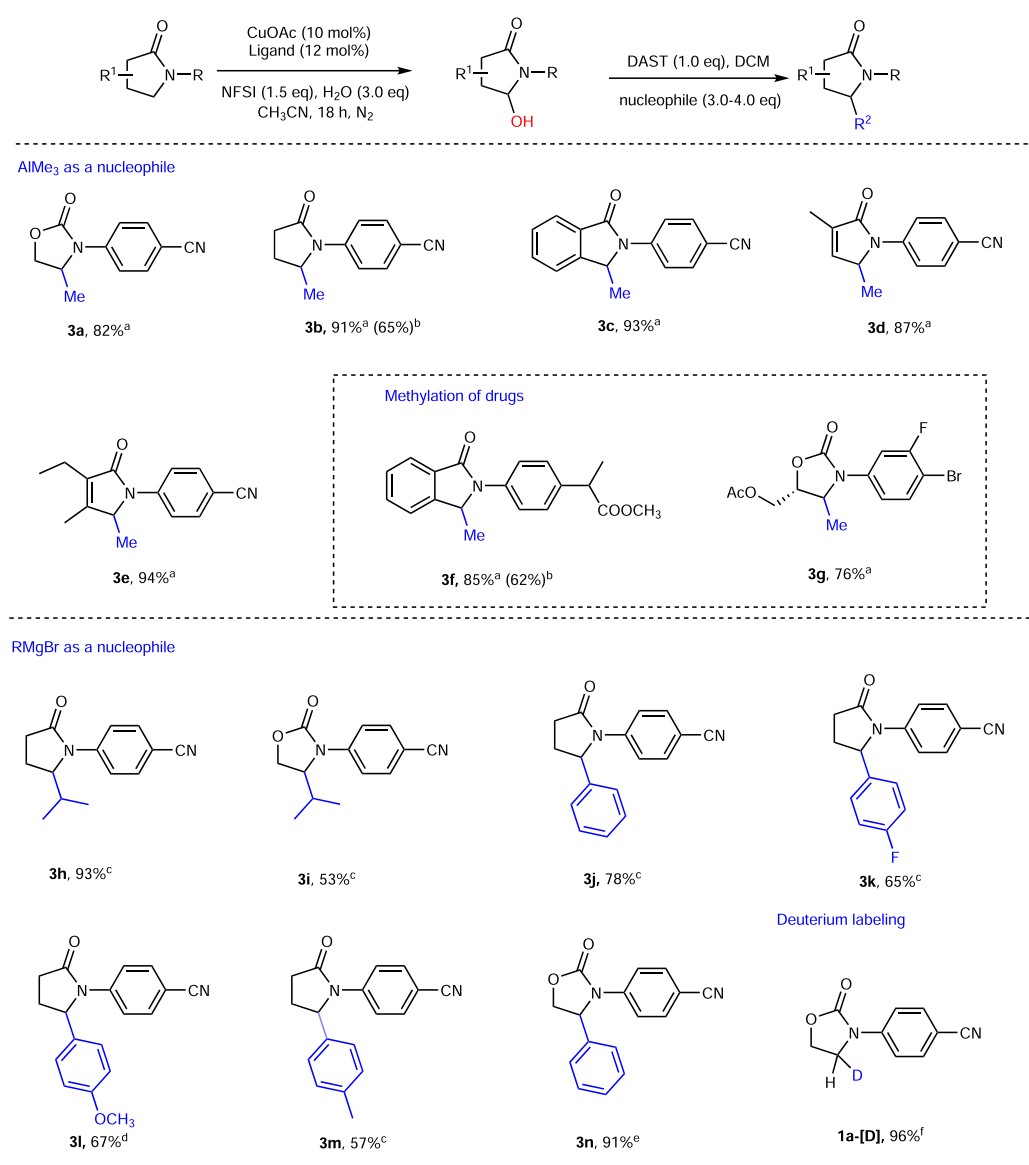

**Fig. 3 Late-stage alkylations.** Reaction conditions: [a]Hemiaminal **2** (0.1 mmol), CH₂Cl₂ (1.0 mL), DAST (0.1 mmol) added at −78 °C; rt for 1 h; cooled to −78 °C, AlMe₃ (0.3 mmol) added, stirred 2 h; rt for 1 h, yield of the second step. [b]**1l** or **1y** (0.2 mmol), H₂O (0.3 mmol), CuOAc (0.02 mmol), **L2** or **L1** (0.022 mmol), NFSI (0.3 mmol), CH₃CN (2.0 ml), under N₂, followed by CH₂Cl₂ (1.0 mL), DAST (0.1 mmol) at −78 °C; rt for 1 h; cooled to −78 °C, AlMe₃ (0.3 mmol) added, stirred 2 h; rt for 1 h, overall yield of two steps. [c]Hemiaminal **2** (0.1 mmol), CH₂Cl₂ (1.0 mL), DAST (0.1 mmol) added at −78 °C; rt for 1 h; cooled to −78 °C, RMgBr (0.4 mmol) added, stirred for 5 h; rt for 1 h. [d](A) bpy (0.4 mmol), THF (0.5 mL), RMgBr (0.4 mmol) added at −40 °C for 0.5 h; rt for 0.5 h; (B) **1l** (0.1 mmol), CH₂Cl₂ (1.0 mL), DAST (0.1 mmol) added at −78 °C; rt for 1 h; (B) added to (A), stirred 5 h at −40 °C; rt for 1 h. [e](A) CuBr·Me₂S (0.4 mmol), THF (0.5 mL), RMgBr (0.4 mmol) added at −40 °C for 0.5 h; rt for 0.5 h; (B) **1a** (0.1 mmol), CH₂Cl₂ (1.0 mL), DAST (0.1 mmol) added at −78 °C; rt for 1 h; (B) added to (A), stirred for 5 h at −40 °C; rt for 1 h. [f]**1a** (1.0 mmol), TFA (2.0 mL), NaBD₄ (4.0 mmol) added at 0 °C; rt for 2 h.

conditions furnishing the CDC products in good yields (**2de–2dg**, 76–84%).

In view of the importance of glycoconjugation of drugs in pharmaceutical chemistry, which has been extensively used to improve the cancer cell selectivity and solubility of drugs in aqueous solution[46–48], two monosaccharide derivatives were tested for C–H glycoconjugation. Both hexose derivative and pentose derivative were suitable substrates for this transformation and excellent yields were observed (**2dh**, 85% and **2di**, 83%). The C–H functionalization reaction also proceeded effectively in generating structural diversity for late-stage functionalization of a number of pharmaceuticals, including the nitrofuran antibiotic-nitrofurantoin (**2z1**, 51%), the precursor of antibiotic-tedizolid (**2z2**, 72%), anti-inflammatory drug-methyl ester of indoprofen

(**2z3**, 73%), and a natural product proline-based dipeptide **2z4** (61%). To demonstrate the scalability of this method, gram-scale reactions of **2a**, **2aa**, **2ia**, **2n**, and **2q** were performed and comparable yields were observed with those of the small-scale reactions (95% vs. 91%, 89% vs. 83%, 86% vs. 81%, 91% vs. 79, and 59% vs. 61%, respectively).

**Synthetic utilities**. Our subsequent studies focused on late-stage alkylation of these heterocycles. The late-stage alkylation includes two sequences of reactions: (1) CDC approach using H₂O as coupling partner to give hemiaminal and (2) Lewis acid or fluorination reagent promoted nucleophilic substitution. As shown in Fig. 3, alkylation reagents such as organoaluminum

reagents and Grignard reagents were tested. All these organo-metallic nucleophiles reacted well with the fluoride intermediates, obtained by treating hemiaminals with diethylaminosulfur tri-fluoride (DAST), to produce the desired products in good yields (53–94%). Given the importance of "magic methyl" effect in drugs[49, 50], we also performed methylation reactions of drug **1z3**

**Fig. 4 Synthetic applications of pyrrolidine derivative 2s.** Reaction conditions: **2s** (0.1 mmol), CH$_2$Cl$_2$ (1.0 mL), BF$_3$·OEt$_2$ (0.2 mmol), and nucleophile (0.3 mmol) added at −78 °C for 1h; rt for 3 h. (a) 1,2-Dimethylindole as a nucleophile and the reaction was performed at −78 °C for 3 h. (b) AlMe$_3$ as a nucleophile. (c) Et$_2$Zn as a nucleophile. (d) 2-Naphthol as a nucleophile. (e) Allyltributylstannane as a nucleophile. (f) TMSN$_3$ as a nucleophile.

and a drug precursor **1z2**, and both of the methylated products **3f** and **3g** were obtained in excellent yields (85% and 76%, respectively). Remarkably, late-stage methylation can also be successfully performed in a one-pot manner without purification of hemiaminals. For example, methylated products **3b** and anti-inflammatory drug **3g** can be accessed via a one-pot procedure in 65% and 62% yield, respectively, which are comparable with those obtained from two steps with purification of intermediates. With Grignard reagents, iPr and substituted phenyl Grignard reagents were used to demonstrate the late-stage alkylation reaction and the desired products were successfully obtained (53–93% yield). Given the importance of deuterated drugs, we also successfully applied this protocol to mono-deuteration of drugs using **1a** as an example (96% yield).

Late-stage functionalization of pyrrolidine derivatives was carried out next, as pyrrolidine is one of the most common N-heterocycle in drugs and peptides[1]. As shown in Fig.4, **2s** as an example, treatment of **2s** with BF$_3$·OEt$_2$ and various organic nucleophiles provided a diverse set of functionalized pyrrolidine derivatives in generally moderate-to-good yields (59–90%). Similarly, the OH group can be successfully replaced by organoaluminum reagents and organozinc reagents to produce 2-Me and 2-Et-substituted pyrrolidines **3p** and **3q** in 90% and 78% yield, respectively. Moreover, **2s** can also be successfully converted to 2-allyl product **3s** and 2-N$_3$ product **3t** by organotin reagent and organosilicon reagents (85% and 76% yield, respectively). In addition, Friedel–Crafts reactions of 1,2-dimethylindole and 2-naphthol with **2s** were also successful affording **3o** and **3r** in 59% and 71% yields, respectively.

**Mechanistic investigations.** Concerning the reaction mechanism, we proposed two possible pathways for this C–H functionalization reaction as shown in Fig. 5. Initially, **L**/Cu(I) complex undergoes single electron transfer to NFSI, to produce a bis (phenyl)sulfonamide radical[36–38]. The bis(phenyl)sulfonamide radical then abstracts a hydrogen atom from C($sp^3$)–H adjacent to nitrogen of compound **1** to generate the corresponding radical **A**. In Path I, a radical chain propagation process is proposed,

**Fig. 5 Proposed reaction mechanism.** Two pathways proposed for this C–H functionalization reaction: fluorination via radical chain mechanism (Path I) and fluorination through a Cu(I)/Cu(II) catalytic cycle (Path II).

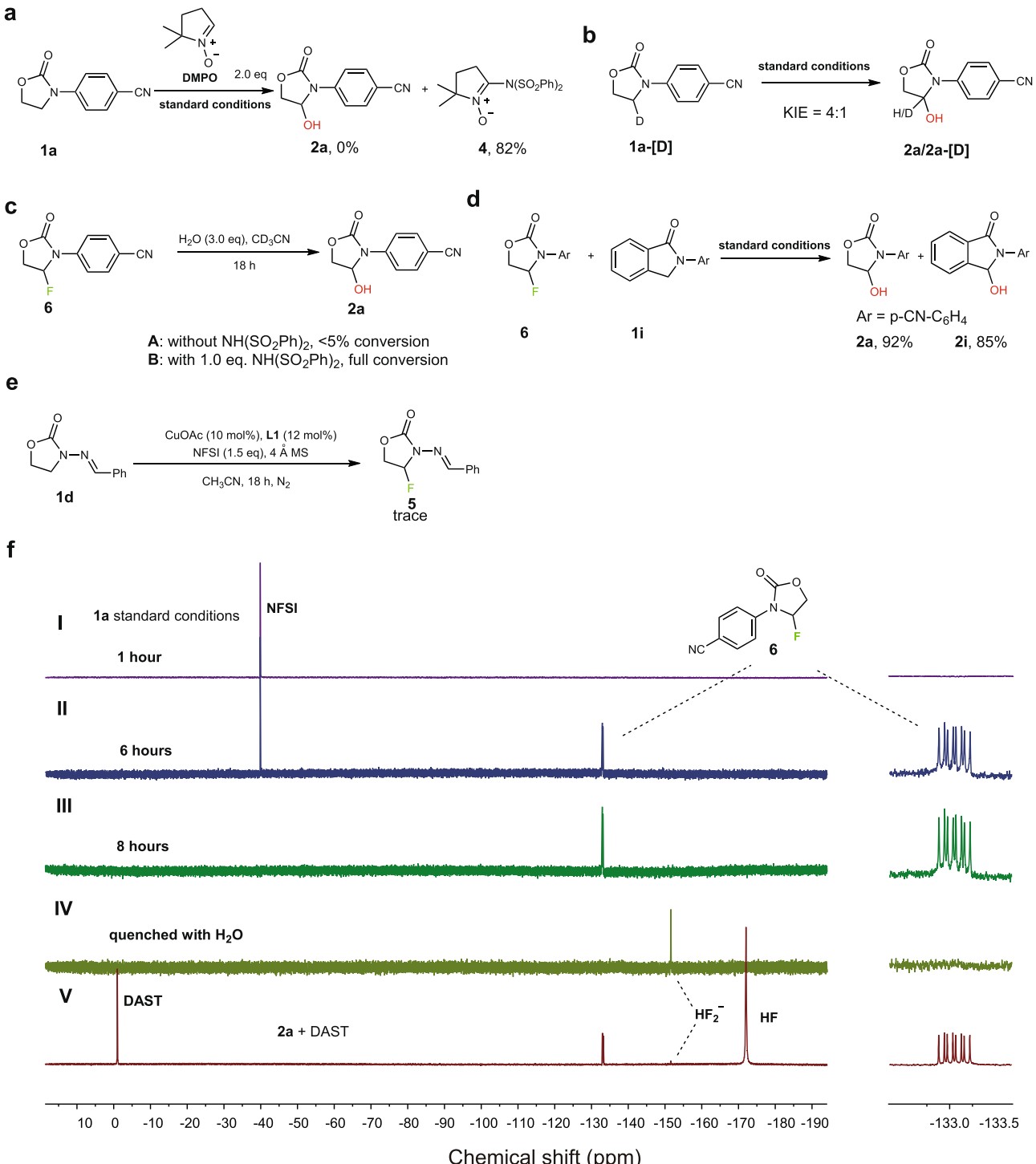

**Fig. 6 Mechanistic experiments. a** Free-radical scavenging experiment. **b** KIE experiment. **c** Conditions: CD$_3$CN (0.5 mL), **6** (0.1 mmol), and H$_2$O (0.3 mmol). **d** Cross-over experiment of the intermediate **6** and **1i** under the standard conditions. **e** Conditions: in glovebox, 4 Å molecular sieve (0.5 g), CH$_3$CN (1.0 mL), **1d** (0.1 mmol), CuOAc (0.01 mmol), ligand **1** (0.012 mmol), and NFSI (0.15 mmol). **f** $^{19}$F (376 MHz) NMR spectra of the N-heterocyclic C–H functionalization reaction. Conditions: CD$_3$CN (0.5 mL), **1a** (0.1 mmol), CuOAc (0.01 mmol), ligand **1** (0.012 mmol), and NFSI (0.15 mmol). (I) $^{19}$F (376 MHz) NMR spectrum of the N-heterocyclic C–H functionalization reaction after 1 h. (II) $^{19}$F (376 MHz) NMR spectrum of the reaction after 6 h. (III) $^{19}$F (376 MHz) NMR spectrum of the reaction after 8 h. (IV) $^{19}$F (376 MHz) NMR spectrum of the reaction after adding 0.01 mL H$_2$O. (V) Fluorinated intermediate **6** prepared from **2a** and DAST: CD$_3$CN (0.5 mL), **2a** (0.1 mmol), and DAST (0.1 mmol) .$^{19}$F (376 MHz) NMR spectrum of the reaction after 1 h.

which is supported by recent work on alkyl radical fluorination[51, 52], especially the work on fluorine transfer to alkyl radical from N–F reagents[51]. The radical **A** undergoes radical fluorination with NFSI to generate the corresponding fluorination intermediate **B** with regeneration of the bis(phenyl)sulfonamide radical to enter a radical chain process. In Path II, a Cu(I)/Cu(II) cycle is proposed. The radical **A** undergoes radical fluorination with Cu$^{II}$-F species by bimolecular homolytic substitution (SH$_2$)

reaction to generate the fluorination intermediate **B**[53, 54] with regeneration of **L**/Cu(I) complex. Finally, the fluorinated intermediate **B** undergoes a nucleophilic substitution promoted by $NH(SO_2Ph)_2$ via the iminium intermediate **C** to afford the final CDC product **2** with elimination of HF[54].

To verify our hypothesis, free-radical scavenging experiments with 5,5-dimethyl-1-pyrroline N-oxide (DMPO) was performed (Fig. 6a). The result showed that by adding 2.0 equiv. of DMPO into the reaction of **1a** under standard reaction conditions, the C–H hydroxylation reaction was completely inhibited and no trapped adduct from DMPO and radical **A** was observed. Instead, the initially formed bis (phenyl)sulfonamide radical was successfully trapped by DMPO giving nitroxide radical derivative, which was further oxidized to more stable nitrone **4**[55, 56]. In addition, kinetic isotope effect (KIE) studies were also performed to test the key hydrogen atom transfer (HAT) step (Fig. 6b). The intramolecular H/D competition experiment (KIE: 4 : 1) indicated that HAT is the product-determining step. These results suggested a free-radical process was likely involved in this reaction. To further test our hypothesis, [19]F nuclear magnetic resonance (NMR) spectroscopy experiments were performed to verify the existence of fluorination intermediate[53] and the results were shown in Fig. 6f. When **L1** (0.012 mmol), CuOAc (0.01 mmol), **1a** (0.1 mmol), and NFSI (0.1 mmol) were mixed in $CD_3CN$ (0.5 mL) at room temperature under $N_2$ and kept for 1 h, only the signal of NFSI (−39.4 p.p.m.) was found. After 6 h, a new peak at −133.1 p.p.m. (ddd, $J = 70.0, 35.2, 23.3$ Hz) emerged, which was attributed to fluorination intermediate **6**. The structure of the intermediate was unambiguously confirmed by an authentic sample prepared from **2a** and DAST in $CD_3CN$ (Fig. 6f, V). Identical chemical shifts, peak patterns, and coupling constants in both [19]F and [1]H NMR spectroscopy were observed in these samples (Fig. 6f, II, III, and V, and Supplementary Fig. 2). After 8 h, full consumption of NFSI was achieved and the signal of $HF_2^-$ (−151.6 p.p.m.)[57] was observed after the reaction was quenched by adding excess $H_2O$. The hydroxylated product **2a** and fluorination intermediate were both clearly seen in [1]H NMR spectrum and ~50% conversion was achieved (Supplementary Fig. 2). To further confirm the role of $NH(SO_2Ph)_2$, the hydrolysis of **6** was studied and the conversion is followed by [1]H NMR. When the hydrolysis of **6** with $H_2O$ (3.0 eq.) was performed in $CD_3CN$ under neutral conditions, the reaction is very slow and only <5% conversion was observed after 18 h (Fig. 6c). However, in the presence of $NH(SO_2Ph)_2$ (1.0 eq.), **6** was completely converted into **2a** after 18 h (Fig. 6c). Next, to further confirm that the fluorination intermediate can be transformed into the final product under the reaction conditions, a cross-over control experiment was performed. The result showed that adding fluorination intermediate **6** prepared individually to the reaction of **1i** under the standard reaction conditions did not affect the reaction and the corresponding hydroxylated product **2a** and **2i** were both obtained in good yield. (Fig. 6d). Altogether, these experiments clearly proved the existence of fluorination intermediate during the reaction process.

Further investigations revealed an interesting finding for the mechanism of this reaction: specifically, the presence of $H_2O$ can substantially improve the rate of fluorination process. When the reaction of **1d** was carried out under dry conditions in the presence of 4 Å molecular sieves, low conversion of the reaction was observed and only trace amount of fluorinated intermediate **6** was detected by [19]F NMR (Fig. 6e). Furthermore, when the reaction of **1d** and MeOH (3.0 eq.) was performed under dry conditions in the presence of activated 4 Å molecular sieves, low conversion was observed as well. It is likely that $H_2O$ molecules interact with fluorine by H-bonding interactions, which possibly facilitates the radical fluorination process[58, 59].

## Discussion

In summary, we have developed a Cu(I)-catalyzed late-stage C ($sp^3$) α-functionalization of N-heterocycles. A series of N-heterocycles, including oxazolidinones, isoindolinones, lactams, pyrrolones, pyrrolidines, piperidines, quinolones, and imidazolidine-2,4-diones, was applicable to the present methodology. The protocol includes two sequences of reactions as follows: CDC reaction using $H_2O$ or alcohol as a coupling partner and subsequent Lewis acid or fluorination reagent promoted nucleophilic substitution. The intermediates hemiaminal or aminal are stable and can be isolated and used for various synthetic purposes. A broad range of nucleophiles were suitable for the present late-stage functionalization of these heterocycles including four drugs. Furthermore, this protocol is also applicable to glycoconjugation of drugs as exemplified by two monosaccharides derivatives, as well as mono-deuteration of drugs. Preliminary mechanistic studies reveal that the reaction undergoes a C–H fluorination process, which is followed by a substitution by $H_2O$ or alcohol to give stable hemiaminal or aminal products. Besides, it was found that $H_2O$ molecules could substantially accelerate the rate of fluorination process by H-bonding interactions with fluorine. This protocol features mild reaction conditions, high chemoselectivity, broad scope of both substrates and nucleophiles, and commercially available catalysts, which ensure its practical application in late-stage α-functionalization of N-heterocycles in natural products and drugs.

## Methods

**General procedure for C($sp^3$)–H functionalization of N-heterocyclics**. In a Schlenk tube, N-heterocyclic compounds **1a** (0.2 mmol), NFSI (0.3 mmol), **L1** (0.024 mmol, 12 mol%), and CuOAc (0.02 mmol, 10 mol%) were dissolved in $CH_3CN$ (1.0 mL) under a $N_2$ atmosphere, then $H_2O$ (0.6 mmol) was added. The reaction mixture was stirred at 35 °C for 24 h. Upon completion, saturated sodium bicarbonate (30 mL) was added and the reaction was extracted with dichloromethane (DCM) (2 × 30 mL), washed with brine (30 mL), dried over $Na_2SO_4$, and concentrated in v

acuo. The residue was purified by column chromatography on silica gel with a gradient eluent of petroleum ether and ethyl acetate to provide the desired product **2a** in a yield of 89%.

## Data availability

The authors declare that the data supporting the findings of this study are available within the paper and its Supplementary Information file. The experimental procedures and characterization of all new compounds are provided in Supplementary Information. Extra data are available from the corresponding author upon request.

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

## Acknowledgements

We are grateful for the support of this work by the National Natural Science Foundation of China (21801260 and 21971267), Guangdong Provincial Key Laboratory of Chiral

Molecule and Drug Discovery (2019B030301005), the 1000-Talent Youth Program of China, and the program for Guangdong Introducing Innovative and Entrepreneurial Teams (2017ZT07C069).

## Author contributions

D.Z. and Z.C. conceived and designed the experiments and mechanism studies. D.Z. and R.W. guided the project. Z.C., J.H., G.C., H.Z., and W.S. performed the experiments. Z.C. and D.Z. wrote the manuscript. All the authors discussed the results and commented on the manuscript.

## Competing interests

The authors declare no competing interests.
