## [Peer Review File · Nature Communications]

REVIEWER COMMENTS

Reviewer #1 (Remarks to the Author):

This manuscript from Zhao and co-workers describes the development of a copper-catalysed C–H functionalization of alpha-N-heterocycles.

I do not feel this manuscript carries enough novelty for publication in NatComm as it is essentially a different way to promote the same chemistry already developed by the White group (Nature 2020). I am not convinced that the use of Cu vs Mn and NFSI over H₂O₂ is a significant advancement in the area, considering that the substrate scopes are essentially identical. The scope is nice and the following functionalizations are interesting, however we have just seen all of this in the White paper. I think if the authors could show some aspects of novelty especially if this method can address some limitations of the previous ones, then my assessment would be different. Of course the limitations have to be related to the key C–H oxidation step, not the following functionalization.

The authors need also to revise their mechanistic scheme and discussion as it contains many fundamental mistakes (e.g. a transition state is mistaken for an intermediate).

It is very strange that the addition of TEMPO suppresses the reactivity and doesn't lead to the isolation of the TEMPO-adduct.

What is the peak at –170 ppm in figure 6eV?

Reviewer #2 (Remarks to the Author):

A Cu(I) catalyzed late-stage C(sp³)-H functionalization of N-heterocycles via CDC approach was described in this manuscript. Different N-heterocycles and monosaccharides derivatives were compatible in in this strategy. Besides, reaction mechanism was proved by control experiments and intermediate experiments. However, it is necessary to make some revises before publication:
1 Many different alcohols were explored, but I want to could hindered alcohols such as tert-butanol and cyclic alcohols be compatible; Substrates with electron withdrawing groups on aromatic rings are more effective, but what will happen if using substrates with electron donating groups on aromatic rings, over oxidation or other byproducts.

2 Line 15 "We have investigated 8 types of N-heterocycles which usually found as medicinally important skeletons." should be "We have investigated 8 types of N-heterocycles which are usually found as medicinally important skeletons."; Line 158, "Late-stage functionalization of pyrrolidine derivatives was carried out next, as pyrrolidine is one the most common" should be "Late-stage functionalization of pyrrolidine derivatives was carried out next, as pyrrolidine is one of the most common

nitrogen heterocycle in drugs and peptides".

3 These literatures should be cited: "Synthesis, 2019, 51, 83–96"; "Green Chem., 2020, 22, 3742–3747". J. Am. Chem. Soc. 2020, 142, 41, 17693–17702

Reviewer #3 (Remarks to the Author):

The manuscript reported an efficient copper catalyzed late-stage C(sp³)-H functionalization of nitrogen heterocycles under mild conditions with a wide substrate scope. The method can be scaled up and applicable to pharmaceuticals and carbohydrates. Detailed mechanistic studies have also been carried out. The manuscript is well prepared. Overall, the work is acceptable with the following revisions:

1. The ligands used in the reaction are chiral ones, have the authors observed enantioselectivities? These should be included.

2. Ligands with aromatic substituents give much better results than the ones with alkyl substituents. Can the authors give a rationale on what caused these difference.
3. To further confirm the fluoro intermediate, can they authors perform experiment with the pre-formed fluoro compound can the same conditions with the nucleophile? Can you use React IR to follow the reaction to confirm the fluoro being the intermediate.
4. Some steps of the catalytic cycle seem not balanced.
5. Some minor errors need to be corrected: P1, "contains" should be "contain", "despite of" should be "despite". P4: :a slightly modified condition was" or "slightly modified conditions were". P5: "under this condition" or "Under these conditions", add "and" before "n-butyl".

Reviewer #1: We thank the referee for the positive evaluation, such as: The scope is nice and the following functionalizations are interesting,..., as well as careful analysis, reading and constructive suggestions.

Point 1: I do not feel this manuscript carries enough novelty for publication in NatComm as it is essentially a different way to promote the same chemistry already developed by the White group (Nature 2020). I am not convinced that the use of Cu vs Mn and NFSI over H₂O₂ is a significant advancement in the area, considering that the substrate scopes are essentially identical. The scope is nice and the following functionalizations are interesting, however we have just seen all of this in the White paper. I think if the authors could show some aspects of novelty especially if this method can address some limitations of the previous ones, then my assessment would be different. Of course the limitations have to be related to the key C–H oxidation step, not the following functionalization.

We argue that this finding is novel as we show a first example of copper catalyzed late-stage C(sp₃)-H hydroxylation of nitrogen heterocycles via fluorination with broad substrate scope and clear mechanism is shown, which is not known and published. The hemiaminal chemistry is very old and well established, not established by White group. The key breakthrough in White paper is hydroxylation without oxidation to carbonyl.

We agree that Mn/ H₂O₂ is “greener” in terms of green chemistry, but the Mn/ H₂O₂ system needs a chiral ligand which is not commercially available and **ten steps** are required to prepare the active L/Mn catalyst as shown below (figure taken from **J. Am. Chem. Soc. 2013, 135, 14052–14055**). This multistep synthesis is not trivial for most chemists especially for medicinal chemists and weeks will be taken to prepare this catalyst. Whereas those privileged BOX type ligands are commercial and exist in every corner of many organic chemistry labs. Every chemist can reproduce this chemistry at will.

Synthesis and Characterization of C–H Oxidation Catalyst Fe(CF₃-PDP) (3):

REF: **J. Am. Chem. Soc. 2013, 135, 14052–14055; Nature Chemistry 2019, 11, 213–221.**

The key difference between our finding and White paper is that our system undergoes a radical fluorination process whereas Mn chemistry is a direct radical hydroxylation. Our approach is a CDC reaction and it is a completely different approach with different mechanism. This is very important because the fluorinated intermediate can be transformed in situ to many other functional groups in one step. As one example, we can couple almost all alcohols with nitrogen heterocycles as shown in Fig. 2. We can do glycoconjugation of drugs in a single step which is very important for pharmaceutical chemistry as well as chemical

biology. We are very optimistic about future late-stage direct functionalization of nitrogen heterocycles with other functional groups in a single step. Besides, our approach is advantageous over direct hydroxylation using OH radical since hyperconjugative activation of hemiaminals often leads to overoxidation to the carbonyl. As seen in White paper, great care should be taken to avoid overoxidation, eg low temperature, dropwise adding of H₂O₂ etc.

Importantly, regarding to functional group tolerance, our approach can tolerate C=C bonds and secondary alcohol which are not suitable for previous approach due to different mechanism of these two approaches. It is known that the C=C bond can be oxidized to epoxide by H₂O₂, with Mn or Fe catalyst and AcOH (*Chem. Commun.*, 2011, **47**, 4273–4275, *Org. Biomol. Chem.*, 2014, 12, 2062). In order to show that C=C bond is applicable to our system, we have synthesized a new substrate **1h** with a terminal C-C double bond, and performed the reaction under standard conditions, the product **2h** was obtained in high yield without affecting the C=C bond (86%). Secondary alcohols such as isopropanol and cyclohexanol, which are prone to oxidized to ketones with White catalyst, are suitable functional group for our catalysis. Finally, our substrate scope of nitrogen heterocycles is not completely overlapped with White paper, for example, **2d-2h**, **2p-2q**, **2z1** are new substrates.

Point 2: *The authors need also to revise their mechanistic scheme and discussion as it contains many fundamental mistakes (e.g. a transition state is mistaken for an intermediate).*

The proposed catalytic cycle in Figure and description in the text have been revised as requested.

The text has been revised as shown below:

In Path II, the radical **A** reacted with **L**/Cu(II)-F via transition state **TS1** to generate the corresponding fluorination intermediate **C** along with regeneration of the **L**/Cu(I) catalyst⁵¹⁻⁵³. The fluorinated intermediate undergoes a nucleophilic substitution promoted by NH(SO₂Ph)₂ via the iminium intermediate **B** to afford the CDC product **2** with elimination of HF⁵³.

Point 3: *It is very strange that the addition of TEMPO suppresses the reactivity and doesn't lead to the isolation of the TEMPO-adduct.*

We found that TEMPO reacted with NFSI even without copper catalyst. TEMPO was reduced to the corresponding amine without trapped adduct observed. So we used DMPO instead which was also used to trap radical intermediate of NFSI type F-reagent previously by Liu group (ref. 55). Since the bis(phenyl)sulfonamide radical was the radical species initially formed, the bis(phenyl)sulfonamide radical reacted with DMPO to give a nitroxide radical which was further oxidized to stable adduct nitron **4**. The product nitron **4** was isolated and characterized by ¹H NMR, ¹³C NMR and HRMS (SI. Page 48, 139).

Point 4: *What is the peak at -170 ppm in figure 6eV?*

Thanks for your kind reminding. The peak at -151 ppm is assigned to HF₂⁻ instead of HF. The peak -170 ppm is assigned to HF. The chemical shift of HF indeed varies remarkably with solvents, concentration of HF as well as the presence other compounds which can form

H-bonds with HF, due to aggregation of HF molecules. However, the chemical shift of HF₂⁻ is very stable.

Reviewer #2: *We thank the referee for the very positive evaluation, careful reading and kind suggestions.*

Few minor changes requested:

Point 1: *Many different alcohols were explored, but I want to could hindered alcohols such as tert-butanol and cyclic alcohols be compatible;*

Thanks for your kind suggestions. A cyclic substrate cyclohexanol is included in Fig 2, which is applicable to this protocol affording the CDC products **2dd** in 82% yield. However, tert-butanol is too hindered to react with this method and no product was observed.

Point 2: *Substrates with electron withdrawing groups on aromatic rings are more effective, but what will happen if using substrates with electron donating groups on aromatic rings, over oxidation or other byproducts.*

Two new substrates **2m** and **2r** with electron donating groups on aromatic rings were synthesized and proved to be applicable (Fig 2, marked in red color, 90%, 56%). However, substrates with phenyl ring or electron donating groups on aromatic rings, those with similar structure with oxazolidinone **1a** and lactam **1n**, were indeed also studied and proved to be less effective. The reactions were sluggish at low temperature and underwent both over oxidation and elimination at high temperature.

Point 3: *Line 15 “We have investigated 8 types of N-heterocycles which usually found as medicinally important skeletons.” should be “We have investigated 8 types of N-heterocycles which are usually found as medicinally important skeletons.”; Line 158, “Late-stage functionalization of pyrrolidine derivatives was carried out next, as pyrrolidine is one the most common” should be “Late-stage functionalization of pyrrolidine derivatives was carried out next, as pyrrolidine is one of the most common nitrogen heterocycle in drugs and peptides”.*

Many thanks to your carefully reading. These errors have been corrected.

Point 4: These literatures should be cited: “Synthesis, 2019, 51, 83–96”; “Green Chem., 2020, 22, 3742–3747”. J. Am. Chem. Soc. 2020, 142, 41, 17693–17702

These literatures have been included as requested (ref 22-24).

Reviewer #3 *We thank the referee for the very positive comment, careful reading and kind suggestions about confirmation of intermediate!*

Few minor changes requested:

Point 1: *The ligands used in the reaction are chiral ones, have the authors observed enantioselectivities? These should be included.*

This is a good suggestion. If the fluorinated intermediate is enantiomeric enriched and the substitution of alcohol is SN_2 , we should observe enantioselectivities. We have indeed measured the ee of **2d** but it is racemic. Details about chiral separation is also included in the SI (page 27). Besides, for **2dh**, **2di** where enantiomerically pure sugars are used, only 1:1 dr was observed.

Point 2: *Ligands with aromatic substituents give much better results than the ones with alkyl substituents. Can the authors give a rationale on what caused these difference.*

These experiments have been repeated several times. Similar phenomenon is also observed in a previous report by Liu group (*Science* **353**, 2016, 1014-1018), where R= H, 25% yield; R= tBu, 6% yield. It is believed in a latter report (*Nature* 2019, **574**, 516-521) that bis(phenyl)sulfonamide radical binds to copper center via [-S=O...Cu], so it is highly likely that the aromatic substituent of the ligand might interact with aromatic ring of bis(phenyl)sulfonamide radical through Pi-Pi stacking to stabilize the transition state of [-S=O...Cu-F] and facilitate the reaction. Still, it is hard to get firm conclusion.

Point 3: *To further confirm the fluoro intermediate, can they authors perform experiment with the pre-formed fluoro compound can the same conditions with the nucleophile? Can you use React IR to follow the reaction to confirm the fluoro being the intermediate.*

We have performed a cross-over experiment as shown in Fig 6e. When fluorination intermediate **6** was added to the reaction of **1i** under standard reaction conditions, both of them are transformed to the corresponding hydroxylated product **2a** and **2i** in high yields. We have also monitored the reaction process with React IR as suggested and unambiguously confirmed the fluorinated species being the intermediate as shown in Supplementary Fig 3 (page 10). We can see clearly the formation and decay of fluoro intermediate.

A few sentences were added as well:

*Next, to further confirm that the fluorination intermediate can be transformed into the final product under the reaction conditions, a cross-over control experiment was performed. The result showed that adding fluorination intermediate **6** prepared individually to the reaction of **1i** under the*

standard reaction conditions didn't affect the reaction and the corresponding hydroxylated product 2a and 2i were both obtained in good yield. (Fig. 6e).

Point 4: *Some steps of the catalytic cycle seem not balanced.*

The proposed catalytic cycle has been revised as requested to avoid confusion.

Point 5: *Some minor errors need to be corrected: P1, "contains" should be "contain", "despite of" should be "despite". P4: :a slightly modified condition was" or "slightly modified conditions were". P5: "under this condition" or "Under these conditions", add "and" before "n-butyl".*

Many thanks to your carefully reading. These errors have been corrected.

REVIEWER COMMENTS

Reviewer #1 (Remarks to the Author):

The authors have done a good job at spelling out the key differences between this approach and the one from White. I am now more positive about the novelty of this work. However, it is crucial that they include this discussion in the text so readers might also appreciate it.

There are however many outstanding issues related to the mechanism that has not been addressed and that needs absolutely to be corrected.

1) TS1 is not a transition state! In the actual form, I do not know what it is as it does not make any sense. I strongly advise the authors to remove it completely from the scheme since they clearly do not understand it and furthermore the exact mechanism of radical fluorination by metal-F species can go by S_H2 reaction (as the authors are trying but fail to represent here) or by radical capture by the metal followed by reductive elimination.

2) The idea of a copper-catalytic cycle is reasonable but another and more likely option is that Cu(I) generates the (PhSO₂)₂N• by SET with NFSI and that triggers a radical chain propagation where the (PhSO₂)₂N• undergoes HAT with 1a and radical A generates C by radical fluorination with NFSI. I strongly believe this to be more likely than the copper cycle since the radical fluorination of alpha-N-radical is well precedented. Figure 5 and the discussion need to be revised considering this much more likely option.

3) Since the authors observe the formation of C by ¹⁹F NMR I wonder if the discussion about the formation of B in Figure 5 is pertinent at all. I would strongly advise the authors to remove it.

4) The authors have done a lot of work on the mechanistic aspects of these reactions but I am afraid many of these reactions offer little understanding. For example, Figure 6D does not really show much and should be removed. The experiment in Figure 6E also is of little help. What is important would be to make compound 6 and to expose it to H₂O or MeOH with and without (PhSO₂)NH to see if it evolves into 2a or the product with the OMe. Copper and NFSI do not need to be present.

In conclusion, I am more supportive about this work but major revisions regarding the mechanistic part are absolutely required or the work might be very misleading and fundamentally flawed.

Reviewer #2: Only made remarks to the editor and recommended acceptance of the paper.

Reviewer #3 (Remarks to the Author):

The authors have made revisions to the original manuscript to make it more comprehensible, and more detailed mechanistic studies were conducted. All my concerns have been addressed, as a result, I recommend the publication of this manuscript in Nature Communication.

Reviewer #1: *We thank the referee for the positive evaluation of the manuscript this time, as well as discussion and constructive suggestions about the mechanism.*

Point 1: *The authors have done a good job at spelling out the key differences between this approach and the one from White. I am now more positive about the novelty of this work. However, it is crucial that they include this discussion in the text so readers might also appreciate it.*

Thanks for your appreciation of the novelty of this work and these discussions have been included in the main text.

Point 2: *There are however many outstanding issues related to the mechanism that has not been addressed and that needs absolutely to be corrected.*

1) *TS1 is not a transition state! In the actual form, I do not know what it is as it does not make any sense. I strongly advise the authors to remove it completely from the scheme since they clearly do not understand it and furthermore the exact mechanism of radical fluorination by metal–F species can go by SH2 reaction (as the authors are trying but fail to represent here) or by radical capture by the metal followed by reductive elimination.*

2) *The idea of a copper-catalytic cycle is reasonable but another and more likely option is that Cu(I) generates the $(\text{PhSO}_2)_2\text{N}^\bullet$ by SET with NFSI and that triggers a radical chain propagation where the $(\text{PhSO}_2)_2\text{N}^\bullet$ undergoes HAT with 1a and radical A generates C by radical fluorination with NFSI. I strongly believe this to be more likely than the copper cycle since the radical fluorination of alpha-N-radical is well precedented. Figure 5 and the discussion need to be revised considering this much more likely option.*

We fully agree with you that radical chain process could well be the pathway of the C-F bond formation step. The radical chain pathway has been included.

Point 3: *3) Since the authors observe the formation of C by ^{19}F NMR I wonder if the discussion about the formation of B in Figure 5 is pertinent at all. I would strongly advise the authors to remove it.*

This is just a proposed mechanism before control experiments. In fact, we have always observed fluorinated intermediate accompanied by hydroxylated product based on ^1H NMR and never observed fluorinated intermediate alone. So, this experiment does not exclude iminium intermediate formed from oxidation the radical directly because the hydroxylated product could also be obtained from rapid hydrolysis of the iminium intermediate. Since the radical chain mechanism is the most likely pathway, this iminium intermediate pathway from copper cycle is removed.

Point 4: *4) The authors have done a lot of work on the mechanistic aspects of this reactions but I am afraid many of these reactions offer little understanding. For example, Figure 6D does not really show much and should be removed. The experiment in Figure 6E also is of little help. What is important would be to make compound 6 and to expose it to H_2O or MeOH with and without $(\text{PhSO}_2)\text{NH}$ to see if it evolves into 2a or the product with the OMe. Copper and NFSI do not need to be present.*

Thanks for your suggestion. Fig 6D has been removed. Fig 6E (6d in the new version) is recommended by Referee 3. The hydrolysis of compound **6** with or without (PhSO₂)NH has been indeed studied but not included in the previous version. The hydrolysis progress of **6** is monitored by ¹H NMR in CD₃CN at rt. The hydrolysis of **6** (purified by flash column to remove HF and acidic byproduct) in neutral condition is very slow with <5% conversion after 18 h whereas full conv is observed in the presence of 1 eq of (PhSO₂)NH after 18 h. This is now included as Fig 6c.

REVIEWERS' COMMENTS

Reviewer #1 (Remarks to the Author):

The authors have addressed my previous points.